# Evaluation of WRF/Chem model (v3.9.1.1) real-time air quality forecasts over the Eastern Mediterranean

George K. Georgiou[1], Theodoros Christoudias[1], Yiannis Proestos[1], Jonilda Kushta[1], Michael Pikridas[1], Jean Sciare[1], Chrysanthos Savvides[2], and Jos Lelieveld[3,1]

[1]Climate & Atmosphere Research Centre, The Cyprus Institute, Nicosia, Cyprus
[2]Department of Labour Inspection, Ministry of Labour, Welfare and Social Insurance, Nicosia, Cyprus
[3]Atmospheric Chemistry Department, Max Planck Institute for Chemistry, Mainz, Germany

*Correspondence to:* George K. Georgiou (g.georgiou@cyi.ac.cy)

**Abstract.**

We describe and evaluate a high-resolution real-time air quality forecast system over the Eastern Mediterranean, based on a regional, on-line coupled atmospheric chemistry and aerosol model. The WRF/Chem model is used to perform daily, 3-day forecasts of regulated pollutants ($NO_2$, $O_3$, PM2.5) over the Eastern Mediterranean, applying three nested domains with horizontal resolutions of 50km, 10km and 2km, the latter focusing on Cyprus. Natural (dust, sea-salt, biogenic) emissions are calculated online, while anthropogenic emissions are based on the EDGAR-HTAP global emission inventory. A high spatial (1km) and temporal (hourly) anthropogenic emission inventory is used for the island of Cyprus in the innermost domain. The model skill in forecasting the concentrations of atmospheric pollutants is evaluated using measurements from a network of nine ground stations in Cyprus and compared with the forecasting skill of the EU Copernicus Atmosphere Monitoring Service - CAMS. The forecast of surface temperature, pressure, and wind speed is found to be accurate, with minor discrepancies between the modelled and observed 10m wind speed at mountainous and coastal sites attributed to the limited representation of the complex topography of Cyprus. Compared to CAMS, the WRF/Chem model predicts with higher accuracy the $NO_2$ mixing ratios at the residential site with a normalized mean bias of 7% during winter and -44% during summer, whereas the corresponding biases for CAMS are -81% and -84%. Due to the high temporal resolution of the anthropogenic emission inventory, the WRF/Chem model captures more accurately the diurnal profiles of $NO_2$ and $O_3$ mixing ratios at the residential site. Background PM2.5 concentrations influenced by long-range transport are overestimated by the WRF/Chem model during winter (NMB = 54%) whereas the corresponding NMB for CAMS is 11%. Our results support the adoption of regional, on-line coupled air quality models over chemical transport models for real-time air quality forecasts.

## 1  Introduction

The term air quality is used to describe to what extent the troposphere is contaminated with atmospheric pollutants. High concentrations of atmospheric pollutants can be hazardous to human health. The atmospheric pollutants with the strongest evidence for public health concern, include ozone ($O_3$), nitrogen dioxide ($NO_2$), and particulate matter (PM) (World Health Organization, 2018, (visited on 2020-01-19)). Tropospheric ozone is linked to numerous harmful health effects including reduced lung

function, increased frequency of respiratory symptoms, and development of asthma (Lippmann, 1989; Broeckaert et al., 1999; Brunekreef and Holgate, 2002), while $NO_2$ is associated with the development of emphysema-like lesions (Wegmann et al., 2005). Human exposure to particulate matter is associated with acute cardiovascular events and atherosclerosis while they also affect the cardiovascular system directly by entering into the systemic circulation (Kampa and Castanas, 2008).

The Eastern Mediterranean and the Middle East (EMME) region is characterized by high background tropospheric ozone concentrations (Lelieveld et al., 2002, 2009; Zanis et al., 2014), since it is affected by polluted air masses from various sources such as the eastern and central Europe, and the Middle East(Lelieveld et al., 2002; Gerasopoulos et al., 2005; Ladstätter-Weißenmayer et al., 2007; Kalabokas et al., 2008; Kanakidou et al., 2011). The island of Cyprus is located in the eastern part of the Mediterranean Sea, adjacent to the Middle East and North Africa. Cyprus is one of the countries in the EMME
region that faces challenges with the exceedance of air quality limits and compliance with European regulatory standards. For the year 2017, Cyprus reported the second highest mean annual $NO_2$ concentrations (13.7 $\mu g/m^3$) among 41 European countries (European Environment Agency, 2019). The exposure in high $NO_2$ concentrations in Cyprus is estimated to cause about 240 premature deaths per year. This translates to 0.02% of the population which is among the highest of the 41 European countries (European Environment Agency, 2019). In addition, exposure to PM2.5 is responsible for 473 years of life lost
(YLL) per 100,000 inhabitants in Cyprus (European Environment Agency, 2019).

The effects of increased concentrations of air pollution on human health highlight the need for real-time air quality forecasting (RT-AQF) with detail in space and time. RT-AQFs can provide the environmental authorities and the general public with information and warning in advance in order to make informed decisions and take actions that will better protect the population from imminent air pollution episodes.

RT-AQF over the Eastern Mediterranean and Cyprus is provided by the Whole Atmosphere Community Climate Model - WACCM (Gettelman et al., 2019), and the Copernicus Atmosphere Monitoring Service (CAMS). WACCM is a global Chemical Transport Model (CTM) driven by meteorological fields from the Goddard Earth Observing System, Version 5 (GEOS-5) model. The model provides daily 10-day air quality forecasts on a horizontal resolution of $0.9° \times 1.25°$ and 6-hour time-step starting at 00:00 UTC. CAMS provides daily 4-day-ahead air quality forecasts over Europe on a horizontal resolution
of 0.1°and 1-hour time-step, based on an ensemble of 9 state-of-the-art numerical air quality models developed in Europe: CHIMERE from INERIS (France), EMEP from MET Norway (Norway), EURAD-IM from Jülich IEK (Germany), LOTOS-EUROS from KNMI and TNO (Netherlands), MATCH from SMHI (Sweden), MOCAGE from METEO-FRANCE (France), SILAM from FMI (Finland), DEHM from Aarhus University (Denmark), and GEM-AQ from IEP-NRI (Poland).

Cyprus is a small country with steep changes in altitude, and urban centres with radial extent of less than 10km. Georgiou
et al. (2017) have shown that increasing the horizontal model resolution from 80km to 16km, there was an improvement in the simulation of the near-surface concentrations of atmospheric pollutants over Cyprus. No further improvement was shown when increasing the horizontal model resolution to 4km. Similar results were obtained by Kushta et al. (2018). This study attributed this model behaviour to the coarse resolution of the anthropogenic emission inventory. A number of studies have also reported that the quality of the input data strongly affects the performance of the models (Abdallah et al., 2016; Werner et al., 2018;
Im et al., 2018). Furthermore, Georgiou et al. (2020) showed that the implementation of an up-to-date, high spatiotemporal

resolution anthropogenic emission inventory resulted in better representation of both the magnitude and the diurnal profiles of the near-surface concentrations of the atmospheric pollutants over Cyprus, especially near the areas with intense anthropogenic activity. The dramatic increase in computational power during the last two decades now allows the use of online coupled models for high resolution air quality forecasting.

In this work we describe and evaluate a high-resolution RT-AQF system established in the Eastern Mediterranean and Cyprus, based on a regional, on-line coupled air quality model and using a high spatiotemporal resolution anthropogenic emission inventory. We evaluate the skill of the RT-AQF system to forecast the atmospheric concentrations of $NO_2$, $O_3$, and $PM2.5$, which are the three regulated by the European Union atmospheric pollutants with the strongest evidence for their effects on human health  (World Health Organization, 2018, (visited on 2020-01-19)). Regional air quality models are able to

run in very high horizontal resolutions, down to the convection permitting resolution limit of the order of 1 km. This allows for more accurate representation of the topography and the population density, and therefore anthropogenic emissions, which is important in a small country like Cyprus with steep changes in altitude, and urban centres with radial extent of less than 10km.

The manuscript is structured as follows: In Section 2 we describe the domain set-up, the model configuration, and the model input data. We examine the skillfulness of the WRF/Chem model to forecast the basic meteorological parameters (Section 3.1),

the concentrations of $NO_2$ (Section 3.2), $O_3$ (Section 3.3), and $PM2.5$ (Section 3.4). Our conclusions are given in Section 4.

## 2   WRF-Chem model and observations

### 2.1   Model configuration

We use the Weather Research and Forecasting model with Chemistry (WRF/Chem) version 3.9.1.1 to perform 3-day-ahead meteorological and air quality forecasts starting at 00:00 LST every day for the three winter (January, February, December) and

three summer (June, July, August) months of 2020. We use three one-way nested domains (Fig. 1) with horizontal resolutions of 50 km, 10 km, and 2 km. The outermost domain (d01) includes the Black Sea region, the largest part of Europe, and the Middle East and North Africa deserts which have an important contribution to the background concentrations of gas-phase and aerosol pollutants over the EMME region. The second domain (d02) focuses over the Eastern Mediterranean and includes the major urban centres in the Middle East. The third domain (d03) is focused over the island of Cyprus. We use 33 vertical layers,

while adaptive time-stepping is used in order to meet the Courant-Friedrichs-Lewy (CFL) stability criterion at each time-step (Jacobson, 2005) and to reduce the simulation times.

In order to have optimal representation of the meteorological fields, for our set-up we are using physics parameterizations which have been optimized for Cyprus by the Department of Meteorology for weather forecast and as well as physics parameterizations that have been shown by Zittis et al. (2014) to have the better performance in terms of simulating several

meteorological variables including total precipitation and air temperature over the Eastern Mediterranean and the Middle East region.

Various gas-phase chemistry and aerosol mechanisms have different behavior in terms of predicting the atmospheric concentrations of pollutants over specific regions (Gupta and Mohan, 2015; Balzarini et al., 2015; Im et al., 2015; Mar et al., 2016).

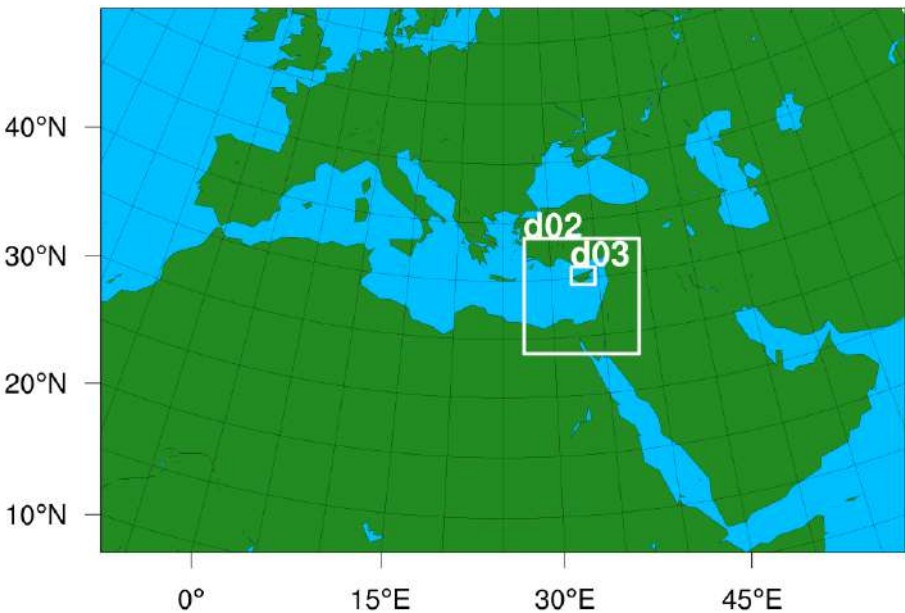

**Figure 1.** Model simulation domains

Georgiou et al. (2017) evaluated the performance of three different gas-phase chemistry mechanisms in terms of simulating the concentrations of atmospheric pollutants over the Eastern Mediterranean and Cyprus. The Regional Acid Deposition Model, version 2 (RADM2) mechanism was found to have better performance compared to the CMBZ and MOZART mechanisms. The differences between the three mechanisms were attributed to the different reaction rates and the way the mechanisms

treat the volatile organic compounds. Compared to RADM2, RACM includes updated rate constants, based laboratory measurements. In this set-up we use the Regional Atmospheric Chemistry Mechanism (RACM) (Stockwell et al., 1997) which is an updated version of RADM2. Aerosol inorganic species are simulated using the the Modal Aerosol Dynamics Model for Europe (MADE) (Ackermann et al., 1998), while the secondary organic aerosols parameterization based on the volatility basis set (VBS) by Ahmadov et al. (2012) is used for secondary organic aerosols. Detailed description of the model physics

configuration is given in Table 1.

Initial and boundary conditions for meteorology are provided by the Global Forecast System (GFS) every three hours on a horizontal grid resolution of $0.25° \times 0.25°$ (06:00 initialization time). Boundary conditions for the gas-phase species and aerosols are provided by the Whole Atmosphere Community Climate Model (WACCM; Gettelman et al. (2019)). The WACCM model output datasets are available on a horizontal grid resolution of $1° \times 1°$ and interpolated in space every six hours to our

model domain. The dust component from the boundary conditions is not taken into account in our simulations, since the Middle Eastern and North Africa deserts are included in our model domain. A model spin-up time of 3 days was used for the first forecast of each period, while restart files are used for the rest of the forecasts.

**Table 1.** Gas-phase chemistry, aerosols, and physics parameterizations used in the simulations.

| Process | Scheme |
| --- | --- |
| Gas-phase chemistry | RACM (Stockwell et al., 1997) |
| Aerosols | MADE/VBS (Ackermann et al., 1998; Ahmadov et al., 2012) |
| Cloud microphysics | Morrison double moment (Morrison et al., 2005) |
| Longwave & Shortwave radiation | RRTMG (Mlawer et al., 1997) |
| Cumulus parameterization | Grell 3D (Grell, 2002) |
| Photolysis | Fast-J |
| Land-surface physics | Noah Land Surface Model (Chen and Dudhia, 2001) |
| Planetary Boundary Layer | Yonsei University (Hong et al., 2006) |

Biogenic emissions are generated on-line by the WRF/Chem model based on weather and land use data, using the Model of Emissions of Gases and Aerosols from Nature version 2.1 (MEGAN2.1) by Guenther et al. (2012). Dust emissions are simulated using the Georgia Tech/Goddard Global Ozone Chemistry Aerosol Radiation and Transport (GOCART) model (Ginoux et al., 2001). The dust emission flux $F_p$ in the GOCART model is calculated as:

$$F_p = CSs_p u_{10m}^2 \left(u_{10m} - u_t\right) \tag{1}$$

where $S$ is a source function which defines the potential dust source regions, $s_p$ is the fraction of each size class of dust, $u_{10m}$ is the horizontal wind speed at 10m, and $u_t$ is the threshold velocity above which dust emission occur. $C$ is an empirical proportionality constant originally set equal to $1\,\mu gs^2m^{-5}$. The value of $C$ estimated by Ginoux et al. (2001) was initially based on regional data over North America. Zhao et al. (2010) evaluated the performance of the WRF/Chem model for different values of $C$. They found that for $C = 0.4\,\mu gs^2m^{-5}$, the WRF/Chem simulated mean AOD over the Sahel region was consistent with measurements from the Dust and Biomass burning Experiment (DABEX) campaign. Therefore, as that is the most prevalent source of dust emissions in the EMME region, in the following simulations, a value of $C$ equal to $0.4\,\mu gs^2m^{-5}$ is used.

Anthropogenic emissions for the first and second domain of the simulations are based on the EDGAR-HTAP Version 2 emission inventory (Janssens-Maenhout et al.) and were interpolated in time and space to produce daily emissions for the first and second domain using the anthro_emiss utility (Kumar, 2018, (visited on 2020-01-19). For the innermost domain a high-resolution emission inventory developed by Georgiou et al. (2020) is used. This emission inventory uses the total reported emissions of CO, $NO_x$, NMVOC, $SO_2$, and PM for the year 2013 on a $1\,km \times 1\,km$ resolution which is upscaled to the resolution of the innermost domain of the simulations ($2\,km$) using a nearest-neighbour grid-point attribution algorithm, while diurnal, weekly, and monthly emission cycles are applied to each species according to Schaap et al. (2005) and the predominant emission activity per season.

There are two operational power generation stations in the southern part of Cyprus. $NO_x$ emissions from these two stations account for about 27% of the total $NO_x$ emitted from the part of the island which controlled by the Republic of Cyprus. At these locations, the emission factors for power generation are applied for all species. Two additional power generation

**Table 2.** Monitoring stations.

| Monitoring site | Abbreviation | Type of zone | Alt. (m) | Measurements |
|---|---|---|---|---|
| Ayia Marina | AYMBGR | Background | 532 | $T2$, $WS_{10}$, $PSFC$, $CO$, $NO_2$, $O_3$, $SO_2$, $PM2.5$, $PM10$ |
| Larnaca Traffic | LATRA | Traffic | 15 | $T2$, $WS_{10}$, $PSFC$, $CO$, $NO_2$, $O_3$, $SO_2$, $PM2.5$, $PM10$ |
| Limassol Traffic | LIMTRA | Traffic | 19 | $T2$, $WS_{10}$, $PSFC$, $CO$, $NO_2$, $O_3$, $SO_2$, $PM2.5$, $PM10$ |
| Mari Industrial | MARIND | Industrial | 88 | $T2$, $WS_{10}$, $PSFC$, $CO$, $NO_2$, $O_3$, $SO_2$, $PM2.5$, $PM10$ |
| Nicosia Residential | NICRES | Residential | 208 | $T2$, $WS_{10}$, $PSFC$, $NO_2$, $O_3$, $SO_2$, $PM2.5$, $PM10$ |
| Nicosia Traffic | NICTRA | Traffic | 176 | $T2$, $WS_{10}$, $PSFC$, $CO$, $NO_2$, $O_3$, $SO_2$, $PM2.5$, $PM10$ |
| Paphos Traffic | PAFTRA | Traffic | 40 | $T2$, $WS_{10}$, $PSFC$, $CO$, $NO_2$, $O_3$, $SO_2$, $PM10$ |
| Paralimni Traffic | PARTRA | Traffic | 72 | $T2$, $WS_{10}$, $PSFC$, $CO$, $NO_2$, $O_3$, $SO_2$, $PM2.5$, $PM10$ |
| Zygi Industrial | ZYGIND | Industrial | 9 | $T2$, $WS_{10}$, $PSFC$, $CO$, $NO_2$, $O_3$, $SO_2$, $PM2.5$, $PM10$ |

stations, one of them located very close to the city of Nicosia, are operational in the northern part of the island and are not included in the high-resolution emission inventory. For these areas, the emission inventory takes into account the emissions from the EDGAR-HTAP global emission inventory, resulting in underestimation of the total $NO_x$ emissions. Emissions from road transport dominate the $NO_x$ emissions (47%) while there is also important contribution from industrial processes (19%).

Industrial processes are the main local source of PM (38%) followed by agriculture (19%). Since 2013, no important changes were observed in the $NO_x$ and PM2.5 total emissions. Specifically, PM2.5 emissions increased from 0.97 kt in 2013 to 1.0t kt in 2019, whereas total $NO_x$ emissions decreased from 15.36 kt in 2013 to 14.04 kt in 2019. Georgiou et al. (2020) showed that using the updated emission inventory in hindcasting mode resulted in reduction of the normalized mean bias between the modelled and observed $NO_x$ mixing ratios at the residential sites (from 67% to 29% and from 51% to 10% for the winter

and summer, respectively). In line with this, the overestimation in $O_3$ mixing ratios was reduced from 45% to 28% during the winter and from 25% to 19% during summer. Finally, taking into account the diurnal variability in the emission inventory was found to be crucial for the simulation of the daily profiles of $NO_x$ and $O_3$ at residential sites.

## 2.2 Observational data

The modelled concentrations of the air pollutants from the 1st day of forecast is compared against hourly observational data

from nine air quality monitoring ground stations, provided by the Cyprus Department of Labour Inspection (DLI) for the three winter (January, February, December) and three summer (June, July, August) months of 2020. During these periods, there were no restrictions in place due to the COVID-19 pandemic. The station network consists of background (Figure 2, green circle), residential (Figure 2, cyan circle), traffic (Figure 2, yellow circles), and industrial (Figure 2, red circles) stations which span the southern part of the island of Cyprus and apart from the concentrations of the air pollutants, also measure various

meteorological variables. The characteristics and the monitored pollutants of each station are shown in Table 2.

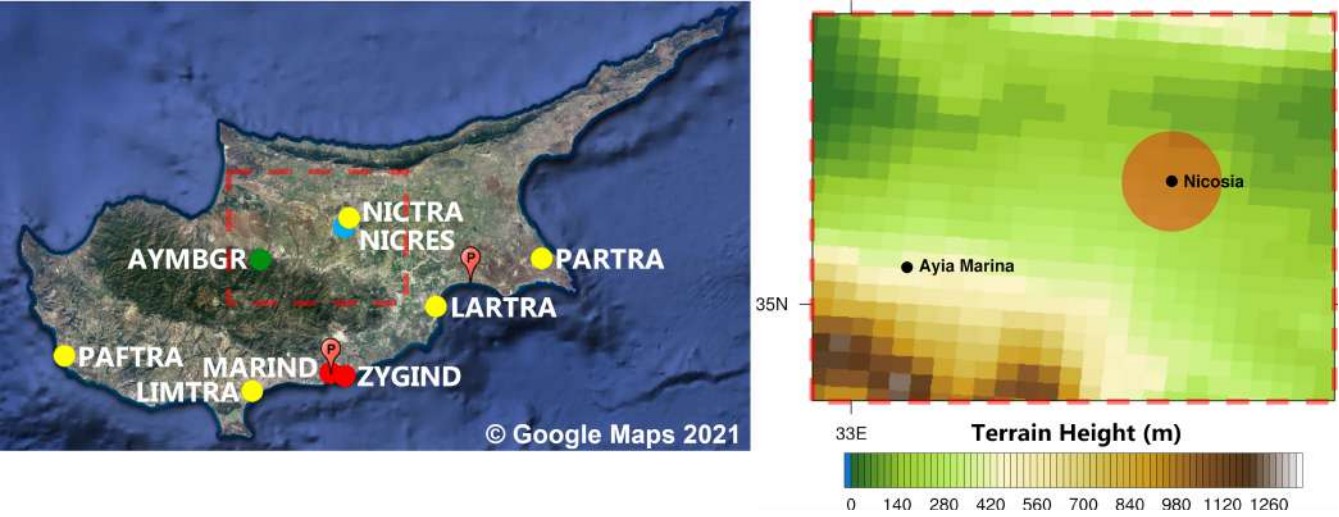

**Figure 2.** Left: Air quality monitoring stations locations (from Google Maps). The the two power generation stations locations are also shown across the south coastline of Cyprus. Right: Terrain elevation and the locations of the Ayia Marina Background station and the city of Nicosia. The red circle indicates the urban area limits.

## 3 Results and discussion

### 3.1 Meteorology

The model skill in forecasting basic meteorological variables is evaluated by comparing the first day of forecast from the WRF/Chem model to hourly measurements from the nine monitoring stations. Table 3 shows the Pearson's correlation coeffi-

5 cient (R), mean bias (MB), normalized mean bias (NMB), and root mean squared error (RMSE) for temperature at 2 m ($T_{2m}$), wind speed at 10 m ($WS_{10}$), and surface pressure ($P_{surf}$) averaged over all stations. Modelled $T_{2m}$ is in good agreement with observations (NMB < 11% for both winter and summer). The diurnal cycle of $T_{2m}$ is also reproduced by the model (R > 0.78) during both seasons. Modelled $P_{surf}$ is in very good agreement with observations with a normalized mean bias of less than 1%. The model tends to overestimate $WS_{10}$ by an average of 2.33 $m/s$ during winter and 1.88 $m/s$ during summer.

Over-predictions of $WS_{10}$ by the WRF model over the Mediterranean have also been reported in previous studies (Zhang et al., 2013; Mar et al., 2016; Georgiou et al., 2017). These biases are mainly attributed to the poor representation of surface drag exerted by the unresolved topography (Zhang et al., 2013). Cheng et al. (2019) reported a decrease in the wind speed overestimation by the WRF/Chem model by increasing the surface roughness length for the urban areas from 0.8m to 2.1m. Therefore, we performed sensitivity tests with increased surface roughness length and examined the effect on the wind speed.

More specifically, surface roughness length for the urban areas has been increased from 0.8m to 2.1m according to Cheng et al. (2019), while forest surface roughness length has been increased from 0.5m to 1m. The increased surface roughness length resulted in a slight decrease of the NMB. The average NMB over all stations was reduced from 144% to 142% during winter and from 101% to 94% during summer.

**Table 3.** Pearson's correlation coefficient (R), mean bias (MB), normalized mean bias (NMB), and root mean squared error (RMSE) between the modelled by the WRF/Chem model and observed hourly values of temperature at 2 m, wind speed at 10 m, and surface pressure averaged over all stations. For all stations during both periods $p - value < 0.05$.

| | Winter | | | Summer | | |
|---|---|---|---|---|---|---|
| | $T_{2m}$ (°C) | $P_{surf}$ (hPa) | $WS_{10}$ (ms$^{-1}$) | $T_{2m}$ (°C) | $P_{surf}$ (hPa) | $WS_{10}$ (ms$^{-1}$) |
| R | 0.80 | 0.61 | 0.37 | 0.78 | 0.88 | 0.52 |
| MB | 1.26 | 3.19 | 2.33 | -1.21 | 2.25 | 1.88 |
| NMB | 0.11 | 0.00 | 1.44 | -0.04 | 0.00 | 1.01 |
| RMSE | 2.94 | 6.63 | 3.33 | 3.08 | 3.35 | 2.64 |

## 3.2 Nitrogen Dioxide (NO$_2$)

The seasonal average NO$_2$ mixing ratios from the observations and the 1st day of the WRF/Chem forecast for winter and summer are shown in Figure 3, 1st row. During both seasons, the higher NO$_2$ mixing ratios appear near the urban centres and the power generation stations (Figure 2). NO$_2$ emitted within the island is shown to affect the eastern part of Cyprus through the prevailing westerly winds which is in agreement with Georgiou et al. (2017).

During both periods, the WRF/Chem model forecasts accurately the background NO$_2$ mixing ratios with a mean bias of less than 1 ppbv at the Ayia Marina background station (Figure 4, 1st row). Similar performance is achieved by the CAMS model. At the Nicosia residential station, the WRF/Chem model outperforms CAMS during winter and summer. The normalized mean bias from the WRF/Chem model is found to be -7% (about 1 ppbv) during winter and -44% (about 3 ppbv) during summer, whereas the corresponding values from CAMS are -81% (about 14 ppbv) and -84% (about 6 ppbv). Underestimation by both WRF/Chem and CAMS is more evident at the traffic stations. The normalized mean bias from the WRF/Chem model is found to be -31% (about 5 ppbv) during winter and -39% (about 4 ppbv) during summer, whereas the corresponding values for CAMS are -86% (about 13 ppbv) and -78% (about 7 ppbv). These stations are located very close to traffic roads and, as a result, they often record very high concentrations of pollutants (as shown by the large number of outliers and the large standard deviation), which cannot be reproduced by the atmospheric model. At the industrial stations, the WRF/Chem model tends to overestimate the seasonal average NO$_2$ mixing ratios by about 4 ppbv during the winter and 7 ppbv during the summer. The biases at these locations can be partly attributed to the fact that in the model, atmospheric pollutants are emitted at the surface, while actual emissions occur at the height of the chimneys which is about 70m above ground. On the other hand, CAMS underestimates NO$_2$ mixing ratios by about 4 ppbv and 5 ppbv during summer and winter respectively. The underestimation of NO$_2$ mixing ratios by CAMS can be attributed to missing sources in the emission inventory used in the models, since it is more evident near locations with intense anthropogenic activity. In addition, the higher resolution of the emission inventory used by the WRF/Chem model results in better representation of the emission sources, and thus, more accurate NO$_2$ forecasts. More specifically, 1km x 1km emissions were upscaled to the innermost domain resolution of 2km while CAMS uses an inventory of a horizontal resolution of 6km x 6km, upscaled to 10km. The statistical metrics for NO$_2$, as well as O$_3$ and PM2.5 at the

background, residential, traffic, and industrial stations are summarized on Table 4. Detailed metrics for each station are given on Table S1 in the Supplement. Similar results were obtained for the second and third day of forecast as shown on Table S2 in the Supplement. This is attributed to the dependency of the model performance on the emissions.

**Table 4.** Pearson's correlation coefficient (R), mean bias (MB), normalized mean bias (NMB), and root mean squared error (RMSE) between the modelled (by the WRF/Chem and CAMS models) and observed hourly values of nitrogen dioxide ($NO_2$), ozone ($O_3$), and fine particulate matter (PM2.5) averaged over the background, residential, traffic, and industrial stations during winter and summer for the first day of forecast.

| | | Winter | | | | | | | | Summer | | | | | | | |
| | | WRF/Chem | | | | CAMS | | | | WRF/Chem | | | | CAMS | | | |
| | | R | MB | NMB | RMSE | R | MB | NMB | RMSE | R | MB | NMB | RMSE | R | MB | NMB | RMSE |
|---|---|---|---|---|---|---|---|---|---|---|---|---|---|---|---|---|---|
| $NO_2$ | Background | 0.12 | 0.62 | 0.36 | 2.18 | 0.39 | -0.89 | -0.52 | 1.25 | 0.03 | -0.42 | -0.45 | 0.85 | -0.15 | -0.57 | -0.61 | 0.91 |
| | Residential | 0.55 | -1.13 | -0.07 | 11.11 | 0.59 | -14.01 | -0.81 | 17.27 | 0.41 | -3.25 | -0.44 | 4.98 | 0.16 | -6.26 | -0.84 | 7.36 |
| (ppbv) | Traffic | 0.36 | -4.87 | -0.31 | 12.11 | 0.46 | -12.83 | -0.86 | 16.22 | 0.16 | -3.79 | -0.39 | 8.36 | 0.23 | -6.54 | -0.78 | 8.71 |
| | Industrial | 0.14 | 4.22 | 0.77 | 12.10 | 0.32 | -3.68 | -0.67 | 5.46 | 0.19 | 6.96 | 0.95 | 15.42 | 0.21 | -5.21 | -0.70 | 7.53 |
| $O_3$ | Background | 0.16 | 2.66 | 0.07 | 10.18 | 0.44 | -3.09 | -0.08 | 6.57 | 0.26 | 2.67 | 0.05 | 9.34 | 0.62 | -10.23 | -0.19 | 12.15 |
| | Residential | 0.49 | 9.91 | 0.48 | 16.55 | 0.65 | 14.41 | 0.70 | 17.37 | 0.40 | 6.58 | 0.15 | 12.33 | 0.67 | -0.93 | -0.02 | 8.24 |
| (ppbv) | Traffic | 0.35 | 12.72 | 0.61 | 18.08 | 0.53 | 15.05 | 0.73 | 18.05 | 0.30 | 10.88 | 0.29 | 15.81 | 0.63 | 6.19 | 0.17 | 10.65 |
| | Industrial | 0.04 | 5.56 | 0.18 | 14.63 | 0.50 | 8.15 | 0.27 | 10.95 | 0.21 | 6.46 | 0.19 | 17.12 | 0.67 | 10.45 | 0.31 | 13.78 |
| PM2.5 | Background | 0.27 | 3.87 | 0.54 | 10.15 | 0.48 | 0.76 | 0.11 | 4.47 | -0.01 | -1.85 | -0.16 | 7.45 | 0.42 | -1.79 | -0.16 | 5.39 |
| | Traffic | 0.19 | -5.01 | -0.28 | 15.19 | 0.32 | -8.76 | -0.50 | 15.02 | -0.04 | -5.44 | -0.32 | 9.51 | 0.59 | -5.54 | -0.33 | 7.39 |
| ($\mu g/m^3$) | Industrial | 0.29 | 1.82 | 0.17 | 9.69 | 0.48 | -1.76 | -0.17 | 5.66 | -0.02 | -2.96 | -0.20 | 8.46 | 0.58 | -3.08 | -0.20 | 5.57 |

Although showing slightly higher Pearson's correlation coefficients between the observed and forecasted $NO_2$ mixing ratios at some of the stations, as shown by the Taylor diagrams (Figure 5, 1st row), CAMS does not capture the diurnal profile of the wintertime $NO_2$ mixing ratios at the locations with intense anthropogenic activity (Figure 6, 1st row). In particular, the forecasted $NO_2$ mixing ratios show very small fluctuations throughout the day, with slight increases during the morning and afternoon hours at the Nicosia residential station and the Larnaca, Limassol, and Nicosia traffic stations. In contrast, the WRF/Chem model captures the peaks which appear in the observations at the residential and traffic stations during the morning and afternoon. These peaks in $NO_2$ mixing ratios appear at the same time with the peaks in the $NO_x$ emissions that have been applied for Cyprus (Georgiou et al., 2020). Due to its short atmospheric lifetime ($\sim$1 day (Seinfeld and Pandis, 2016)), $NO_2$ is detected near the emission sources. The fact that CAMS underestimates the $NO_2$ mixing ratios at the residential and traffic stations but captures the background mixing ratios, indicates the underestimate of local sources in the emission inventory used by the models. Similar results are obtained for the summer period regarding the skill of the models to forecast the diurnal profile of $NO_2$. During the summer, at the residential and traffic stations, lower mixing ratios appear throughout the day, although the $NO_x$ emissions diurnal profile is similar to the winter, with higher emissions during the morning (Georgiou et al., 2020). This can be partly attributed to the boundary layer height (Su et al., 2018; Xiang et al., 2019), (Supplement, Figure S2) and the intense photochemical activity during the summer. These two factors result in a weaker peak in $NO_2$ mixing ratios at the residential station and the majority of the traffic stations. These patterns are resembled by the WRF/Chem model, which

highlights the ability of the model to simulate these characteristics of the concentrations of the atmospheric pollutants near the areas with human presence. The correlation coefficients at the background station are low which is partly attributed to the absence of nearby emission sources. The absence of emissions results in a weak diurnal profile of the concentrations of atmospheric pollutants. Therefore, even small fluctuations in the observed concentrations which are not captured by the model,

result in low correlation between the observed and modelled concentrations. The summertime diurnal profiles of the $NO_2$ mixing ratios are shown in Figure S2 in the supplement.

    The analysis has also been performed for the second domain of the WRF/Chem forecasts with a horizontal resolution of 10km. Compared to CAMS, $NO_2$ forecast from the 10km WRF/Chem domain during winter yields lower NMB for the residential and traffic stations (-70% and -68% compared to -81% and -86% respectively). As mentioned earlier, the NMB is

reduced for the innermost domain of the WRF/Chem (-7% and -31% for the residential and traffic stations respectively). The improvement shown by the innermost (2km) WRF/Chem domain is attributed to the high spatiotemporal resolution emission inventory (Georgiou et al., 2020), and the fact that the utilization of grid spacing closer to the resolution of the emission inventory results in an improvement in the simulation of air pollutants (Kushta et al., 2018). Similar results are obtained for summer highlighting the importance of high resolution air quality forecasts with up-to-date and high spatiotemporal resolution emission

inventories. The statistical metrics from the 10km WRF/Chem domain analysis are given on Table S3 in the Supplement.

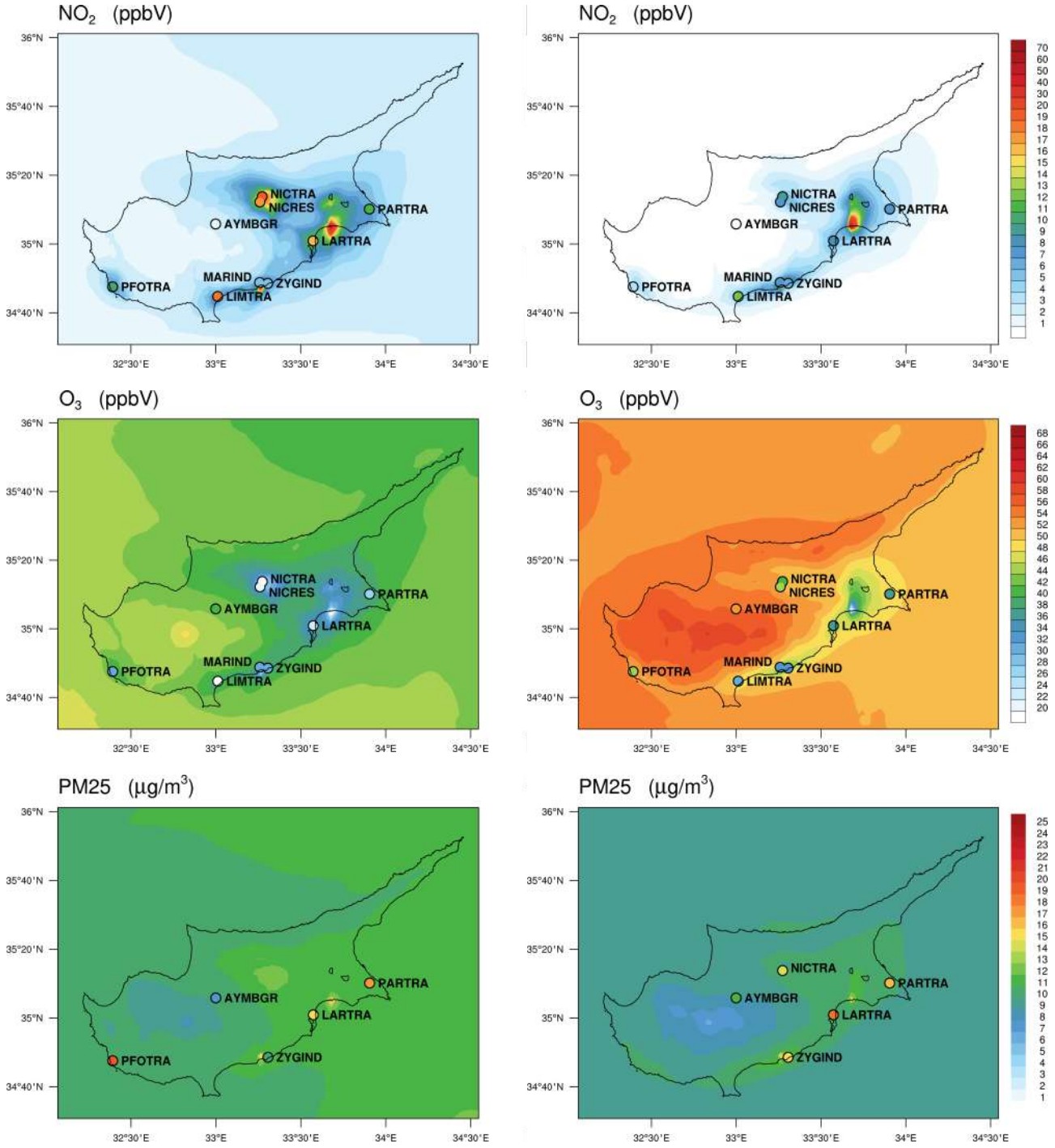

**Figure 3.** Average NO$_2$ (1st row) and O$_3$ (2nd row) mixing ratios, and PM2.5 (3rd row) concentrations from the 1st day of the WRF/Chem forecast during winter (left) and summer (right). The filled dots indicate the observed values.

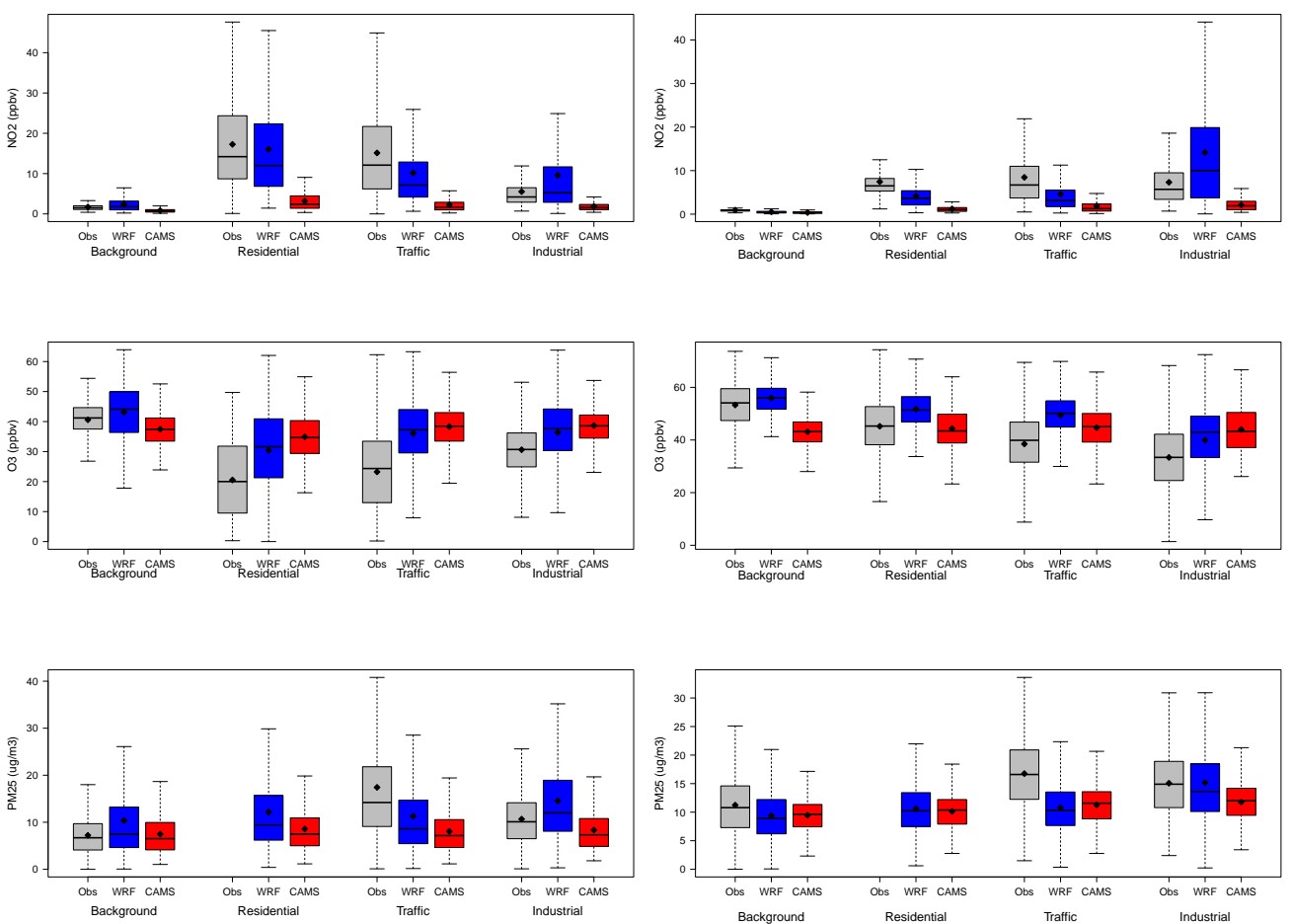

**Figure 4.** Box-and-whisker plots at the background, residential, traffic, and industrial $NO_2$ (1st row), $O_3$ (2nd row), and PM2.5 (3rd row) average mixing ratios in observed data (grey color) and the WRF/Chem (blue color) and CAMS (red color) forecasts during winter (left) and summer (right). The seasonal average mixing ratios are derived from hourly mixing ratios.

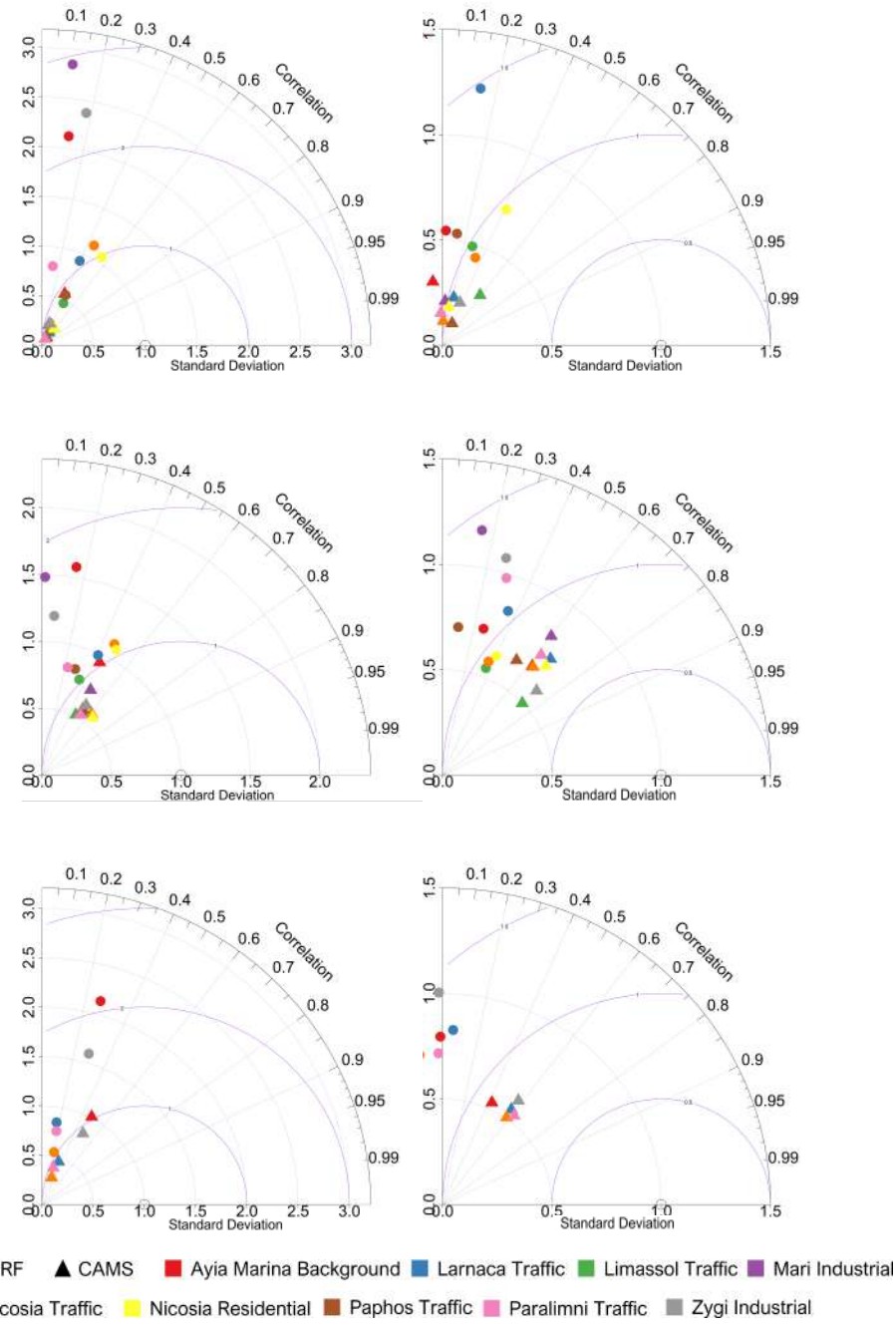

**Figure 5.** Taylor diagrams of $NO_2$ (1st row), $O_3$ (2nd row), and PM2.5 (3rd row) for winter (left) and summer (right).

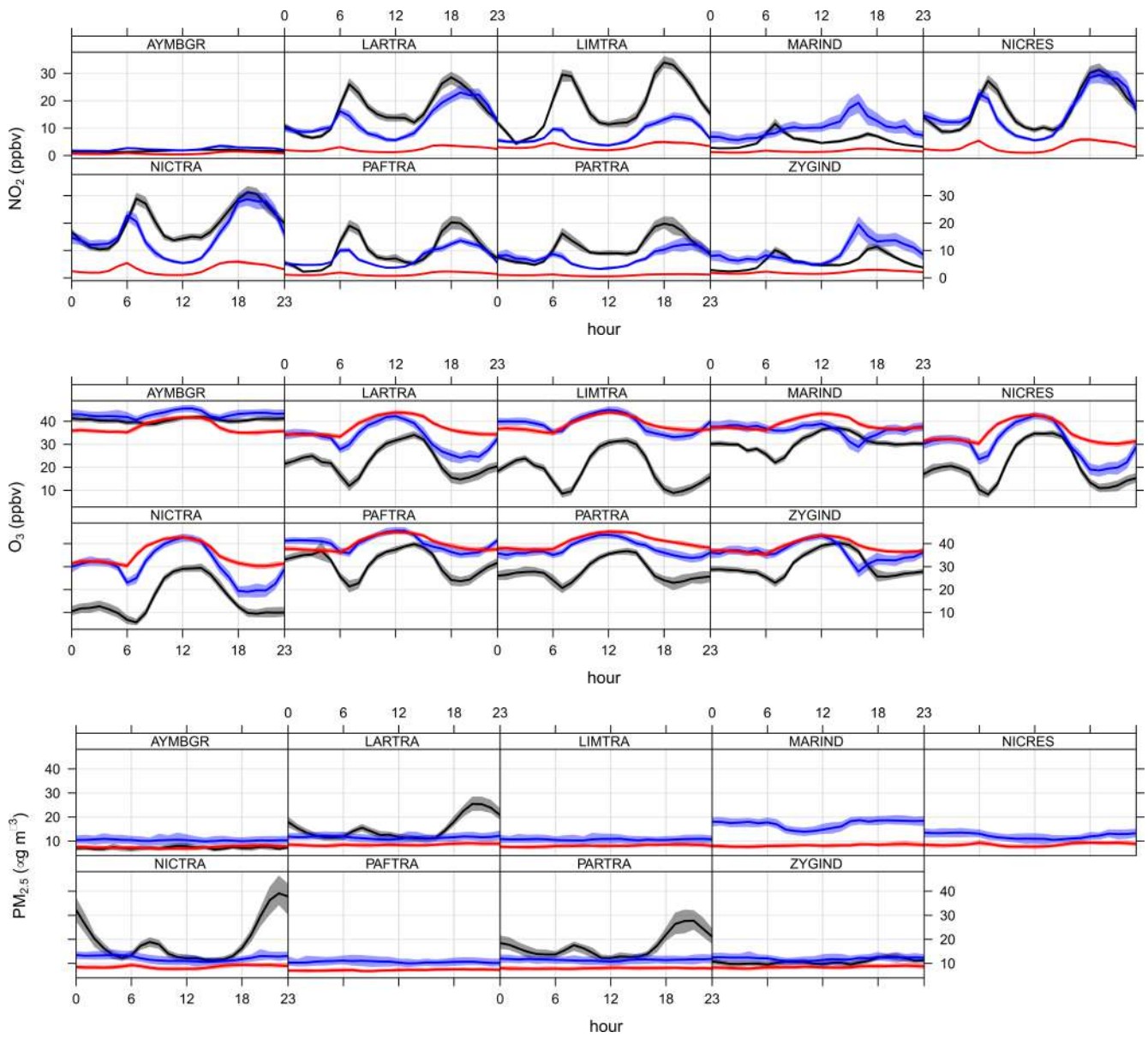

**Figure 6.** Diurnal variation of $NO_2$ (1st row), $O_3$ (2nd row), and PM2.5 (3rd row) in observations (grey lines), the WRF/Chem (blue lines) and the CAMS (red lines) 1st day of forecast during winter.

## 3.3 Ozone (O$_3$)

Observed background O$_3$ mixing ratios reach up to 41 ppbv during winter and 53 ppbv during summer which is in agreement with Gerasopoulos et al. (2005). In their study, using observational data from 1997 to 2004, a maximum in O$_3$ mixing ratios ($58 \pm 10$ ppbv) in July and a minimum ($36 \pm 7$ ppbv) in December were reported over the Eastern Mediterranean. During
both seasons, the lowest O$_3$ mixing ratios appear at the locations with intense anthropogenic activity, such as the locations of the power generation stations and the urban centers, and the eastern part of the island (Figure 3, 1st row), which coincide with the highest NO$_2$ mixing ratios appear (Figure 3, 2nd row). As shown in Figure 4, 2nd row, wintertime background O$_3$ mixing ratios are captured by both WRF/Chem (NMB = 7%) and CAMS (NMB = -8%). Similar performance is shown for the 10km WRF/Chem domain with a NMB of 4%. Summertime background O$_3$ mixing ratios are also captured by the WRF/Chem
(NMB = 5% for both domains), but underestimated by CAMS by about 10 ppbv (NMB = -19%). At the residential and traffic stations, CAMS strongly overestimates O$_3$ mixing ratios during winter by 70% ($\sim$14 ppbv) and 73% ($\sim$15 ppbv) respectively. A strong overestimation is also shown by the 10km WRF/Chem domain at these locations but better performance is achieved by the WRF/Chem high resolution domain where the overestimation is about 48% ($\sim$10 ppbv) at the residential and 61% ($\sim$13 ppbv) at the traffic stations. The overestimation in O$_3$ mixing ratios by both models can be partly attributed at the
underestimation of NO$_2$ mixing ratios seen in Sect. 3.2. During the summer, CAMS underestimates background O$_3$ mixing ratios by about 10 ppbv (NMB = -19%). This results in lower normalized mean bias at the residential and traffic stations (-2% and 17% respectively) compared to WRF/Chem (15% and 29% respectively).

Due to intense photochemical activity and high O$_3$ production, observed O$_3$ mixing ratios show a maximum in the afternoon at the locations with anthropogenic activity. This maximum is successfully forecasted by both WRF/Chem and CAMS (Figure
6, 2nd row). The WRF/Chem model though, is able to forecast with more accuracy the decreases in O$_3$ mixing ratios during the morning and evening hours. These decreases are a result of the increases in NO$_2$ mixing ratios (Sec. 3.2), which are sufficiently forecasted by the WRF/Chem but not by the CAMS model.

### 3.3.1 Ozone daily maximum 8-hour average

Figure 7 shows the O$_3$ daily maximum 8-hour average at the Ayia Marina background station and the urban stations in observa-
tion data, the WRF/Chem and the CAMS forecasts during the summer. During this period, 35 exceedances of the limits set by European Air Quality Directives (EU, 2008, (visited 2022-01-19) have been observed at the Ayia Marina background station, highlighting the effect of the high background concentrations on air quality over the Eastern Mediterranean and Cyprus. In the same period 22 exceedances have been successfully predicted by the WRF/Chem model. CAMS, although predicting more accurately the days with lower O$_3$ daily maximum 8-hour average, did not predict any exceedances of the limit of 60 ppbv
during the whole summer period. This behaviour by CAMS can be partly attributed to the underestimation in summertime O$_3$ background concentrations.

The exceedances of the limit of 60 ppbv at the Nicosia traffic station reached up to 10 during the summer period, while at the other traffic stations, no more than 2 exceedances have been observed. The lower number of exceedances at the traffic stations

can be attributed to the $O_3$ titration by the locally emitted NO. At the Nicosia residential station, 20 exceedances have been observed. Seven exceedances have been successfully predicted by the WRF/Chem model and only two by CAMS. In addition, the WRF/Chem model predicted exceedances that were not shown in observations, as a result of the 15% overestimation in $O_3$ concentrations (Sec. 3.3). During the winter, no exceedances have been observed or predicted due to the lower background $O_3$

5    concentrations over the region.

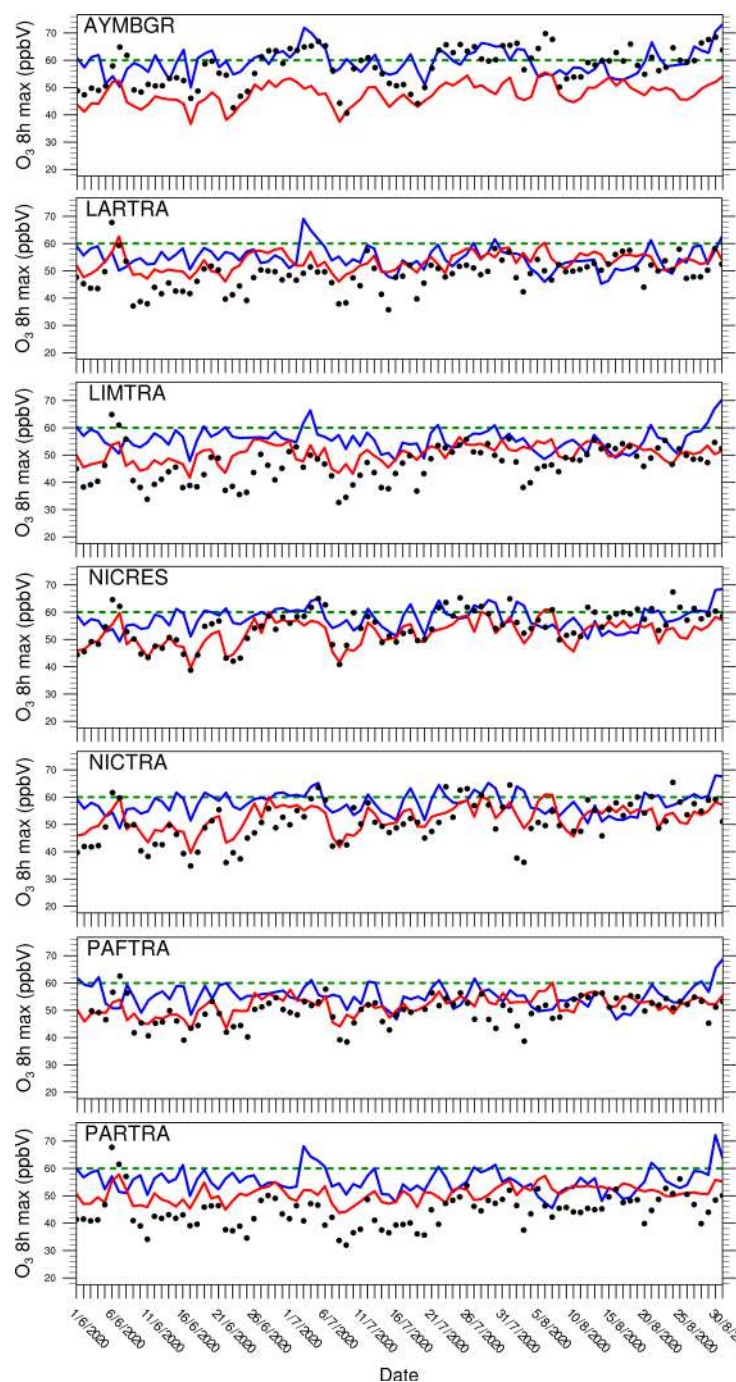

**Figure 7.** O$_3$ daily maximum 8-hour average at the Ayia Marina background station and the urban stations in observation data (black dots), the WRF/Chem (blue lines) and the CAMS (red lines) forecasts during summer. The green line indicates the limit value set by the European Air Quality Directives (EU, 2008, (visited 2022-01-19)).

## 3.4 Fine Particulate Matter (PM2.5)

During summer, observed background PM2.5 concentrations are higher (11.2 $\mu g/m^3$) than winter (7.2 $\mu g/m^3$) which can be partly attributed to the absence of precipitation and the enhanced photochemical conditions which lead to secondary aerosol formation during the summer (Pikridas et al., 2018). The total observed and modelled precipitation for each month is shown on Fig. S3 in the supplement. In addition, the prevalent Etesian winds transport air masses rich in fine mode anthropogenic pollutants from Turkey or to a lesser extent from Europe (Pikridas et al., 2018). Anthropogenic emissions from Turkey and Central Europe were also found by Sciare et al. (2002) to have an important contribution to PM levels over the EMME region. Furthermore, periods with increased levels of non-sea-salt calcium during the summer reported in the study which were associated with air masses Africa and Central Turkey, highlighting the important contribution of background PM2.5 concentrations during the summer.

As can be seen from Figure 4, 3rd row, the WRF/Chem model overestimates background PM2.5 concentrations during winter by about 4 $\mu g/m^3$, while CAMS forecasts wintertime background PM2.5 concentrations with more accuracy ($\leq 1$ $\mu g/m^3$). This overestimation is mainly due to increased PM2.5 concentrations during the 3-day period from December 12th to December 14th which appear at all stations (Figure 8).

At the traffic stations, WRF/Chem and CAMS underestimate PM2.5 concentrations by 5 and 9 $\mu g/m^3$ respectively. During the summer, the two models show similar behaviour. In particular, there is an underestimation of background PM2.5 concentrations of about 16% ($\sim$1.8 $\mu g/m^3$). Underestimation is slightly higher at the industrial stations ($\sim$ 20%). At the traffic stations underestimation reaches up to 5 $\mu g/m^3$ ($\sim$ 33%). The underestimation of PM2.5 concentrations at the traffic stations can in part be attributed to the lack of road dust re-suspension mechanisms in the models and the fact that these stations are located close to main roads. The increased observed PM2.5 concentrations during the evening hours at these stations (Figure 6) are attributed to emission sources from household energy use. These sources are not included in detail in the emission inventory resulting to the underestimation of PM2.5 concentrations during these hours by the model. The low correlation between the modelled and observed background concentrations is partly attributed to the weak diurnal profile of the PM2.5 concentrations and the absence of local emission sources.

To further investigate the WRF/Chem model skill to forecast the PM2.5 concentrations, we examine the contribution of the modelled PM2.5 sub-components to the total PM2.5 concentrations at the Ayia Marina background station and the Nicosia residential station. The hourly concentrations of the PM2.5 sub-components and their contribution to the total PM2.5 concentrations at these stations for winter and summer are shown in Figure 9. The PM2.5 concentrations at both stations are are characterized by increased sulfate aerosol concentrations which is in agreement with Georgiou et al. (2017), as well as increased concentrations of primary PM2.5 aerosols. In the MADE aerosol mechanism, primary PM2.5 aerosols include the emitted accumulation mode dust particles which account for the 7% of the total dust emissions. The important contribution of dust on the total PM concentrations over Cyprus was highlighted by Pikridas et al. (2018). During the period from December 12th to December 14th, increased concentrations of primary PM2.5 are shown. The increased concentrations of primary PM2.5 aerosols appear at both background and residential stations indicates that this overestimation is due to overestimation

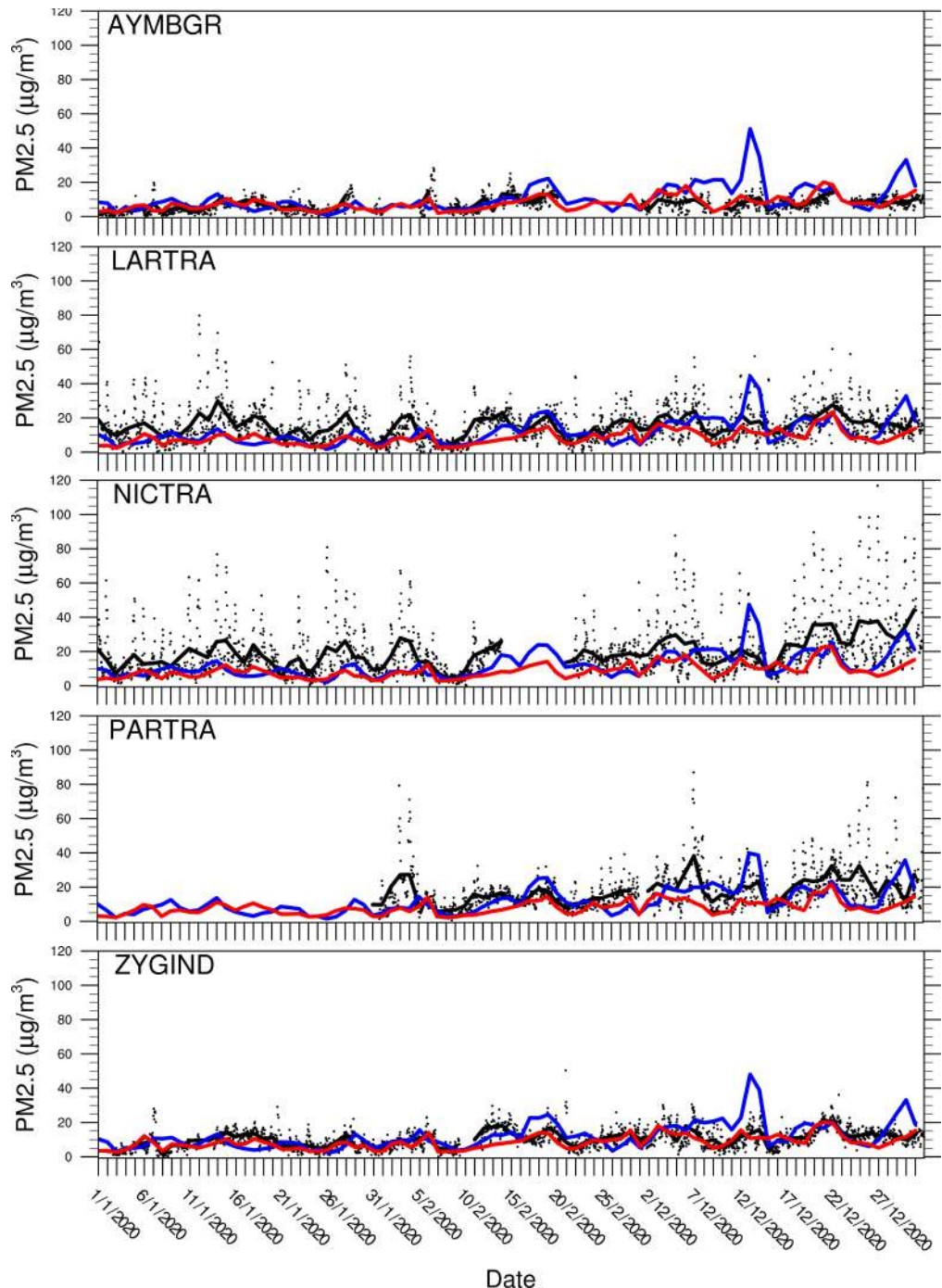

**Figure 8.** Observed hourly PM2.5 concentrations (black dots) and daily average PM2.5 concentrations from observations (black lines), WRF/Chem (blue lines), and CAMS (red lines) during winter.

in dust concentrations. This is also supported by the elevated concentrations of coarse soil-derived aerosols (not shown here) during this period.

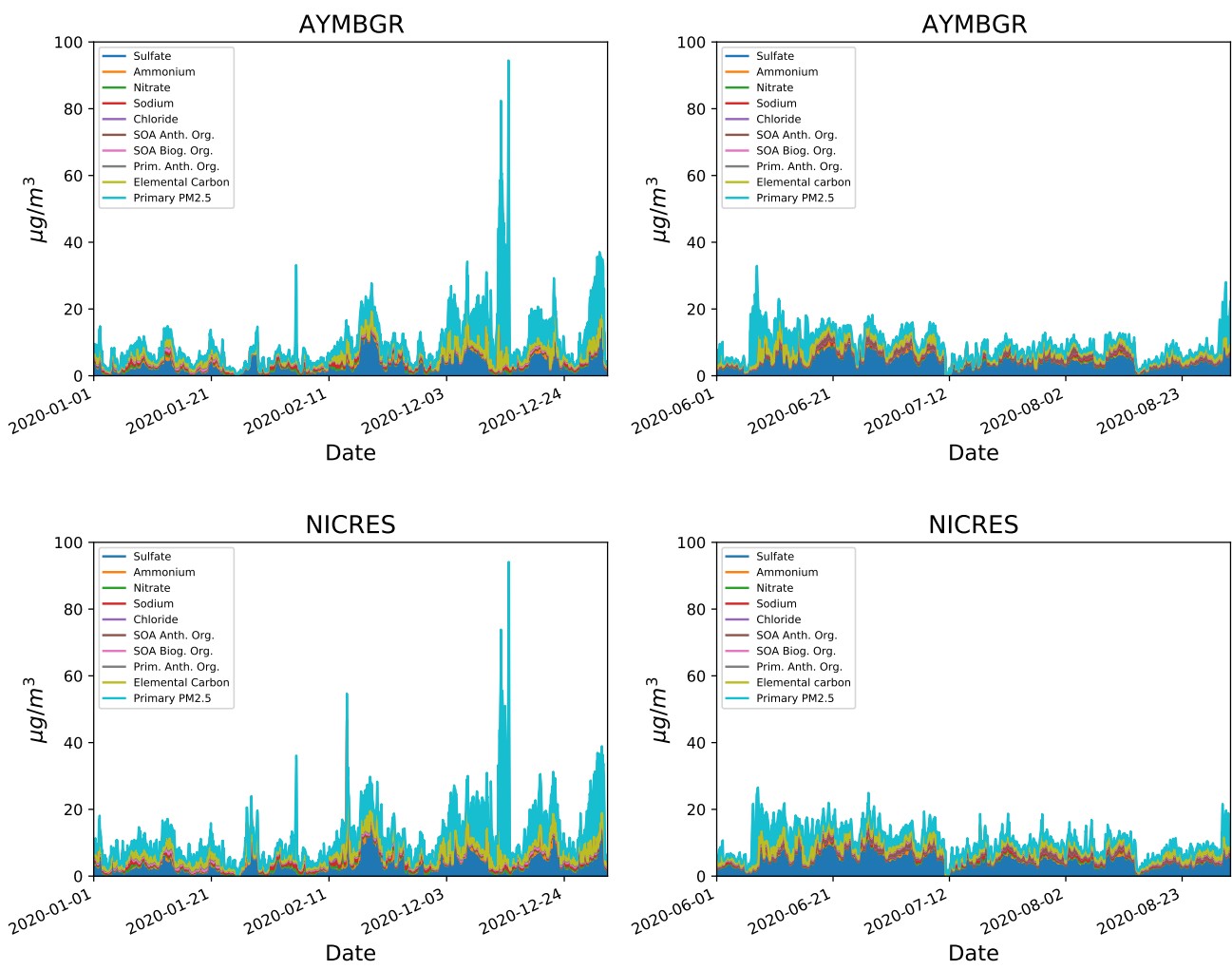

**Figure 9.** Hourly concentrations of the PM2.5 sub-components and their contribution to the total PM2.5 concentrations at the Ayia Marina background station (1st row) and the Nicosia residential station (2nd row) for winter (left) and summer (right).

## 4    Conclusions

The WRF/Chem model has been applied for high-resolution, daily, 3-day-ahead air quality forecasts over the EMME region and Cyprus. The model skill in forecasting the atmospheric concentrations of $NO_2$, $O_3$, and PM2.5, as well as the basic meteorological parameters, has been evaluated for three winter (January, February, December) and three summer (June, July, August) months during 2020. The forecast output is compared against observational data from nine ground stations in Cyprus and forecast data from the state-of-the-art CAMS model. The magnitude and diurnal variation of surface pressure and temperature are accurately forecasted. Wind speed at 10m is overestimated by about 2 $m/s$ which may be attributed to the limited representation of the complex topographical features by the model.

WRF/Chem and CAMS are able to forecast with accuracy the background $NO_2$ mixing ratios during both seasons. $NO_2$ mixing ratios are strongly overestimated by WRF/Chem at the industrial stations which can be partly attributed to the fact that emissions in the model occur at the surface while actual emissions at these locations occur at a height of about 70m. At the residential and traffic stations, WRF/Chem forecasts the magnitude and diurnal variation the $NO_2$ mixing ratios with higher accuracy than CAMS, capturing the morning and afternoon peaks during winter and the morning peak during summer. Consequently, the decreases in $O_3$ mixing ratios caused by the increases in $NO_2$ mixing ratios are successfully forecasted by WRF/Chem. The increased background $O_3$ concentrations during the summer result in a large number of exceedances of the $O_3$ daily maximum average limit at the Ayia Marina background station. A smaller number of exceedances is observed at the urban stations due to the local $NO_x$ emissions. The WRF/Chem model was found to be more skillful in predicting these exceedances, although there were a number of false alarms, attributed to the 15% overestimation of $O_3$ concentrations predicted by the model.

WRF/Chem and CAMS have similar performance in terms of forecasting the PM2.5 concentrations during the summer, revealing an underestimation at the traffic stations which may be attributed to the missing road dust re-suspension mechanism and the proximity of these stations to the main traffic areas. In addition, emissions from domestic energy use which are not accurately represented in the air quality forecasting models, result in underestimation of nighttime PM2.5 concentrations during winter. It is shown that the PM2.5 concentrations in Cyprus are dominated by sulfate aerosols and primary PM2.5 which includes fine dust particles. An overestimation in PM2.5 concentrations is shown by WRF/Chem at the background stations during winter. This overestimation is attributed to increased dust concentrations in the model, suggesting there is potential for further improvement, as dust is a major contributor to PM in the region.

Based on the findings of this study, the use of up-to-date and high spatiotemporal resolution anthropogenic emission inventories and high resolution air quality forecast are recommended for forecasting the concentrations and the diurnal fluctuations of atmospheric pollutants, especially near the urban centers where the majority of population lives. To further improve the forecast capacity of the models, emission heights can be employed for power stations, as well as plume dispersion and chemistry to incorporate the rapid conversion of $SO_2$ to sulphate (adding to the PM2.5 mass).

Coupled on-line air quality models with nesting for higher horizontal resolution can provide improved real-time air-quality forecasts, at least for short-lived species or species that undergo photochemical reactions, compared to the state-of-the-art

global chemical transport models. Adaptive time-stepping can be prescribed in RT-AQF applications with coupled models to reduce the required simulation times while meeting the CFL stability criterion at each time-step. With the advent of efficient processor technology and as computational resources become increasingly available to the modelling community, higher spatial resolutions can be used in regional coupled on-line air quality models which can go down to the convection resolving limit, targeting high population areas; while the use of fine temporal resolution allows for better capturing feedbacks between the meteorological and chemical processes, and diurnal variability.

Finally, a longer evaluation period in future studies covering entire and multiple years can provide a representative larger sample of variability of air-pollution events, and aim to capture extreme but rare events, and the frequent spring and autumn dust episodes in this region. The advances in the predictive skill of real-time air quality forecasting models can help reduce population exposure to air pollutants and associated health risks.

*Code availability.* The WRF source code can be obtained from http://www2.mmm.ucar.edu/wrf/users/download/get_source.html (last access: 27 April 2022) after registration. The modified model code used in this study can be found at this DOI: 10.5281/zenodo.6322996.

*Data availability.* As some of the simulation raw data sets are very large, they can be made available by request from the corresponding author. The processed data used in this study and the post-processing and visualization scripts can be found at this DOI: 10.5281/zenodo.6322996.

*Author contributions.* GKG, TC, JK designed the experiment. GKG carried out the experiment and performed the simulations. GKG, JK and TC analyzed the data and wrote the manuscript. YP provided technical support for implementing the experiment. CS and MP provided and processed the observational data. JS, JL reviewed and edited the manuscript.

*Competing interests.* The authors declare that no competing interests are present.

*Acknowledgements.* The authors are sincerely thankful to the Cyprus Department of Labour Inspection (DLI) for providing the observational data used for the model evaluation in this study, and would also like to acknowledge the Copernicus Atmosphere Monitoring Service (CAMS, https://atmosphere.copernicus.eu/ (visited on 19/01/2022)) which is operated by the European Centre for Medium-Range Weather Forecasts on behalf of the European Commission as part of the Copernicus Programme for the provision of the data used for model intercomparison. The NCAR Command Language (NCL) version 6.5.0 (http://www.ncl.ucar.edu/ (visited on 19/01/2022)) and the openair R package (Carslaw and Ropkins, 2012) were used to produce the plots in this work. This publication has been produced within the framework of the projects EMME-CARE, which has received funding from the European Union's Horizon 2020 Research and Innovation Programme (under grant

agreement no. 856612) and the Cyprus Government, AQ-SERVE no. INTEGRATED/0916/0016 which is co-financed by the European Regional Development Fund and the Republic of Cyprus through the Research and Innovation Foundation, and ACCEPT which is co-financed by the Norwegian Financial Mechanism (85%) and the Republic of Cyprus (15%) in the framework of the programming period 2014–2021.

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
