# Peer review of "Evaluation of WRF/Chem model (v3.9.1.1) real-time air quality forecasts over the Eastern Mediterranean"

_Geoscientific Model Development, 2021_

## Author Response (AR1)

Dear Editor,

We would like to thank the reviewers for the fruitful comments and suggestions that helped us improve the manuscript. Please find our replies to all comments below.

**Anonymous Referee 1**

This manuscript describes the use of an air quality model (WRF-CHEM) with nested domain configuration and a high spatial resolution emission data set for the predictions of air quality on the island of Cyprus. The use of the high spatial resolution emission data set, compared to a global emission data set, improves the forecasting for primary emitted pollutants (e.g., NOx) in urban areas in Cyprus. This is not a particularly novel result in that other studies have shown that a high-resolution emission data set combined with a high-resolution model will yield better results than a global scale model for an urban area.

The use of the high spatial emissions data set is the most significant aspect of this work, and I would encourage the authors to supplement the manuscript with any novel aspects of how the emissions were prepared (e.g., development of spatial surrogate fields from GIS data, population maps, road network maps). The adaptive time stepping is also highlighted as an improvement significantly speeding up the model. Can the authors provide details of the tests performed to optimize the model accuracy with minimal computational cost? This would be of direct interest to readers of GMD.

It appears the goal of the work is to develop an operational forecast model for Cyprus. It is possible that a hybrid model approach could yield with best results with the CAMS model output providing the chemical lateral boundary conditions for the highest resolution WRF-CHEM domain. There may be differences in chemical speciation between models, but it is possible to map lumped species from one mechanism to another. This would be valuable, at least for the longer-lived species, as you might get the best of both methods in the CAMS system capturing global emissions and transport and WRF-CHEM capturing the more local emissions and chemistry. For example, ozone correlation is quite poor within the WRF-CHEM domain compared to the CAMS system (summer R=0.62 vs R=0.26 for background site).

The evaluation considers temperature and wind but does not consider precipitation. Precipitation is an important removal method for soluble gases and particulate matter. Insights can be gained from assessing precipitation and also wet deposition fluxes for pollutants, such as sulfate wet deposition, if observations are available. The wind speed over-prediction seems like a large error in the transport term for local pollutants. A sensitivity test with increased roughness length for mountainous and urban building areas could be performed to assess impact on wind and pollutant concentrations.

In comparing CAMS with WRF-CHEM, it might also be informative to compare the WRF-CHEM 10km domain results with the CAMS results as the spatial resolution would be quite similar and

the authors could isolate differences from long range transport without the effect of grid spacing. I am not suggesting using the 10-km WRF-CHEM results for the operational forecasting but rather as a way to evaluate the outer domain WRF-CHEM model results which are impacted by global emissions and transport.

**General Comments**

- *The emission inventory which is used for operational forecasting is described in detail and evaluated in Georgiou et al., (2020). In this study the high resolution (1km x 1km) emission data which were upscaled to match the innermost domain resolution (2k). Monthly, daily, and hourly emission factors were applied for each emission species according to the predominant activity in Cyprus.*
- *Adaptive time stepping is used in order to meet the Courant-Friedrichs-Lewy (CFL) stability criterion. By using an adaptive timestep, the model automatically decreases the timestep only when needed in order to meet the CFL criterion which is more effective in terms of computational time than using a reduced timestep throughout the whole simulation and thus allow to produce the forecast in shorter wall-clock time.*
- *WACCM is commonly used by the scientific community for both air quality forecasts and research purposes. We choose to use WACCM since the speciation for each gas-phase and aerosol mechanisms are provided by the WRF/Chem developers and are well tested. CAMS provides regional air quality forecasts on a 10km resolution. This domain does not include the main dust sources in the region of interest. These sources are included in the global CAMS air quality forecasts in a horizontal resolution of 40km. We choose though to use an extended domain for the WRF/Chem model with a similar resolution (50km) in which we included the main dust sources in the region as well as the regions from which anthropogenic emissions affect Cyprus such as the Middle East and Europe. This way there is more consistency between the species in the outermost (50km) and second (10km) domains of our simulations.*
- *In order to have as accurate as possible representation of the meteorological fields, for our meteorological set-up, we are using physics parameterizations which have been optimized for Cyprus, applied by the Department of Meteorology for weather forecasting, as well as physics parameterizations that have been shown by Zittis et al., 2014 to have the better performance in terms of simulating several meteorological variables, including total precipitation and air temperature over the Eastern Mediterranean and the Middle East region. We have now added the observed and modelled total monthly precipitation in Figure S3 in the Supplement.*
- *We have performed sensitivity tests in which we increase the surface roughness length for the urban areas from 0.8m to 2.1m according to Cheng et al. (2019), and the forest surface roughness length from 0.5m to 1m. The increased surface roughness length resulted only in a slight decrease of the wind speed NMB. More specifically the NMB was reduced from 144% to 142% during winter and from 101% to 94% during summer. We have included this in our discussion (page7, lines 12-18).*

- *We now include in our analysis (page 10, lines 7-15) the results from the 10km domain of the WRF/Chem model. It is shown that at the residential and traffic stations, $NO_2$ and $O_3$ forecasts from the 10km domain are slightly improved compared to CAMS but biases are still very high compared to the innermost (2k) WRF/Chem domain. This is attributed to the high spatiotemporal resolution emission inventory used for the innermost domain and to the fact that the utilization of grid spacing closer to the resolution of the emission data leads to an improvement in the simulation of air pollutants (Kushta et al., 2018).*

**Specific Comments**

1. **Figure 1.  Red labels are not clear.**

   *We have changed the labels in order to make them clearer.*

2. **Page 4, Line 20.  Duplicate of line 19.**

   *We have removed the line.*

3. **Page 4.  Did the authors consider a spin-up time from the met analysis to develop more higher resolution met fields?**

   *We don't use a spin-up time from the met analysis but rather do a warm start using restart files from the previous WRF/Chem forecast cycle. This applies for both the meteorological and chemical fields.*

4. **Page 5, Line 5. Remove extra period.**

   *The extra period has been removed.*

5. **Figure 5.  Please label sites as urban, rural, industrial, etc.**

   *We have modified the figure legend accordingly.*

**References**

Zittis, G., Hadjinicolaou, P., and Lelieveld, J.: Comparison of WRF Model Physics Parameterizations over the MENA-CORDEX Domain, American Journal of Climate Change, 03, 490–511, doi:10.4236/ajcc.2014.35042, http://www.scirp.org/journal/PaperInformation.aspx? PaperID=52884{&}{#}abstract, 2014.

Kushta, J., Georgiou, G. K., Proestos, Y., Christoudias, T., and Lelieveld, J.: Modelling study of the atmospheric composition over Cyprus, Atmospheric Pollution Research, 9, 257–269, doi:10.1016/j.apr.2017.09.007, https://doi.org/10.1016/j.apr.2017.09.007, 2018.

Cheng, F. Y., Lin, C. F., Wang, Y. T., Tsai, J. L., Tsuang, B. J., and Lin, C. H.: Impact of effective roughness length on mesoscale meteorological simulations over heterogeneous land surfaces in Taiwan, Atmosphere, 10, doi:10.3390/ATMOS10120805, 2019

Georgiou, G. K., Kushta, J., Christoudias, T., Proestos, Y., and Lelieveld, J.: Air quality modelling over the Eastern Mediterranean: Seasonal sensitivity to anthropogenic emissions, Atmos. Environ., 222, 117 119, doi:10.1016/j.atmosenv.2019.117119, 2020.

**Anonymous Referee 2**

The paper compares the forecasting skill of regional high-resolution and real-time WRF/Chem model with EU Copernicus Atmosphere Monitoring Service (CAMS) for regulated pollutants ($NO_2$, $O_3$, $PM_{2.5}$) over the Eastern Mediterranean using 1- and 3-day forecasts and corresponding 9 ground stations in Cyprus. Overall, $NO_2$ forecasts are more accurate for WRF/Chem with normalized mean bias (NMB) of 7% during winter and -44% during summer, whereas the corresponding biases for CAMS are -81% and -84%. For PM forecasts in winter, regional model has NMB of 54% and CAMS has NMB of 11%.

Overall, the paper is useful in verification of regional and global chemical models, but some minor details need to be clarified before the paper should be published. Also, although the paper is interesting and should be published, I was not convinced that WRF-Chem is that much better than CAMS for the air quality forecasts.

**Comments:**

1. **Page 2, Lines 20-25: On those lines it mentions that WACCM provides 10-day forecasts, why these forecasts are not compared to the WRF-Chem and CAMS in the paper?**

   *WACCM provides air quality forecasts in a horizontal resolution of about 90km x 125km which is too coarse to resolve individual stations for model evaluation and meaningful intercomparison. In a previous study (Georgiou et al. 2017) we have compared the output from the WRF/Chem model to the MOZART-4 model at a similar horizontal resolution and we quantified the higher accuracy for the near-surface concentrations of the atmospheric pollutants over the region of interest.*

2. **Page 3, Lines 6-7: "daily 3-day-ahead meteorological and air quality forecasts" – what means daily? Are these 24-hour forecasts?**

   *We perform 3-day ahead forecasts starting at 00:00 every day (i.e., 72 hourly timesteps) We have re-written the sentence in order to make it clear.*

3. **Figure 2: It is not clear how the right map is related to the left map. I recommend expanding the right map or at least placing a city there so it would be possible to find inset right map on the left map.**

   *We have modified the Figure accordingly.*

4. **In Figure 6, is it 1-day or 3-day forecast?**

*Yes, Figure 6 compares observations to the first day of forecast from CAMS and WRF/Chem. We now clarify this in the Figure legend.*

**Minor Comments**

1. **Page 1, Line 23: "(World Health Organization, 2018, (visited on 2020-01-19)" – parentheses are missing.**

   *The missing parentheses have been added.*

2. **Figure 1: There appear to be some red marks next to the drawn domain, but I cannot make out them. Can those markings be made clearer?**

   *We have fixed the labels in order to make them clearer.*

3. **Page 5, Line 4: "There are two operational power generation…" – it feels like it should be a separate paragraph.**

   *We have moved it to a separate paragraph.*

4. **Table 3: Please state between what the correlation is shown. I assume it is between WRF-Chem and ground observations. Same for Table 4.**

   *We have made clear in the captions that the metrics derive from the comparison between the modelled and the observed values.*

5. **Page 7: There are multiple instances of ppbV? Why is V capitalized?**

   *Capitalization has been removed.*

6. **Page 11, Line 25: "(EU, 2008, (visited 2022-01-19)"- parentheses are missing again.**

*The missing parentheses have been added.*

7. **Page 18, Line 18: "False exceedance predictions" – these are usually called false alarms.**

   *We have now adopted the term "false alarms" instead of "false predictions".*

**References**

Georgiou, G. K., Christoudias, T., Proestos, Y., Kushta, J., Hadjinicolaou, P., and Lelieveld, J.: Air quality modelling in the summer over the Eastern Mediterranean using WRF / Chem : Chemistry and aerosol mechanisms intercomparison, Atmospheric Chem. Phys., pp. 1–25, doi:https://doi.org/10.5194/acp-18-1555-2018, 2017.

**Anonymous Referee 3**

**General remarks:**

The paper evaluated a high-resolution real-time air quality forecast system over the Eastern Mediterranean using WRF-Chem. The predicted atmospheric pollutants are evaluated using measurements from a network of nine ground stations in Cyprus and compared with the forecast skill of the EU Copernicus Atmosphere Monitoring Service. It used 3 nested domains and the third domain is 2 km focused over the inland of Cyprus.

The manuscript is not organized well for the model evaluation. The authors only chose NO2, O3 and PM2.5 for evaluation, there is not motivation to explain it. Also, the analysis is jumping here to there without connections. The relationships between NO2 and O3 in the chemical mechanism are not investigated to explain the biases. The PM2.5 evaluation is too simply and need further analysis. What are the major aerosols contributing the PM2.5 consternations? This has not been described. In a lot of places, the conclusions do not have solid evidence without showing the Meteorological Fields analysis, somehow like guess or assumption.

The paper compared the model results with ground-based measurements and the CAMS, but most of these comparisons are only described the biases or differences, there is no solid analysis or investigation to explain what factors attributing to these biases or differences. However, these factors are important to improve the model performance. If the authors would like include other forecast or model data sets (CAMS), more details about the CAMS need to be described, in such a way that the identified differences in the evaluation against observations can be explained.

**General Comments**

- *We chose NO$_2$, O$_3$, and PM2.5 since, according to the World Health Organization, these are the three species regulated by the European Union, being atmospheric pollutants with the strongest evidence for their effects on human health. We have clarified this in the first paragraph of the introduction, providing some examples of the effects of these pollutants on human health as well as the relevant citations.*
- *Various gas-phase chemistry and aerosol mechanisms have different behavior in terms of predicting the atmospheric concentrations of pollutants over specific regions (Gupta and Mohan 2015, Balzarini et al. 2015, Im et al. 2015, Mar et al. 2016). Georgiou et al. (2017), thoroughly investigated the performance of three different gas-phase chemistry and aerosol mechanisms in terms of simulating the concentrations of atmospheric pollutants over the region of interest and Cyprus. The RADM2-MADE/SORGAM mechanism was found to have better performance compared to the CMBZ-MOSAIC and MOZART-MOSAIC mechanisms. More specifically, CBMZ-MOSAIC and MOZART-MOSAIC were found to overestimate the hourly O$_3$ concentrations by 22% and 23% respectively, while the NMB from RADM2 was 9%. RADM2-MADE/SORGAM also achieved better performance in simulating the atmospheric concentrations of NOx, CO, and SO$_2$, as well as the PM2.5. The differences between the three mechanisms were attributed to the different reaction rates*

*and the way the mechanisms treat the volatile organic compounds. More specifically, regarding the reactions directly involving $O_3$, the major differences between RADM2 and CMBZ appear in the inorganic reactions and the treatment of volatile organic compounds, since CMBZ uses a lumped approach for inorganic species. In this work we use the RACM gas-phase chemistry mechanism which is an updated version of RADM2. Compared to RADM2, RACM includes updated rate constants, based laboratory measurements. We now include this in our discussion (page 4, lines 1-7).*

- *Indeed, the contribution of aerosol sub-components to the total PM2.5 concentrations can give insights of the model performance. We now show and discuss the contribution of the PM2.5 sub-components to the total PM2.5 concentrations (page 18, lines 25-34, page 20).*

**Specific comments:**

1. **P2, L20: What is the motivation to mentioned other RT-AQF models? Other than these RT-AQF models, why do you need to use WRF-Chem for regional forecast? Any benefits or priorities to use WRF-Chem compared to these existing RT-AQF models. Better to reorganize the introduction to highlight the motivation of this study**

   *We have reorganized the introduction to more clearly highlight the motivation of this paper. Online coupled models treat meteorological and chemical processes in tandem during the simulations at every timestep. This allows to capture feedbacks between these processes, such as radiation which plays an important role in the production of $O_3$, taking into account all the factors that affect it such as cloud cover and aerosol optical depth. In addition, regional models can run in very high horizontal resolutions, even down to the order of 1km. Previous studies have shown that using a model horizontal resolution closer to the resolution of the emission inventory leads to an improvement in the simulation of air pollutants, but no further improvement can be achieved by just further improving the model spatial resolution. Georgiou et al., (2020) showed that the implementation of an up-to-date, high spatiotemporal resolution anthropogenic emission inventory resulted in better representation of both the magnitude and the diurnal profiles of the near-surface concentrations of the atmospheric pollutants over Cyprus, especially near the areas with intense anthropogenic activity. The dramatic increase in computational power during the last two decades now allows the use of online coupled models for high resolution air quality forecasting. We added a paragraph in the introduction (page 2, lines 29-35 & page 3, lines 1-4) highlighting the motivation of this study, adding the relevant citations.*

2. **P4, L1: Please describe the details about how did you run the model for prediction? What is the forecast length, what is the temporal interval to cycle the chemical Fields, every 6 hours? Is there any chemical initial condition from other model has been included, if not, what it the spin-up time?**

*We perform three-day ahead forecasts every day. The chemical fields boundary conditions are updated every six hours. A spin-up time of three days was used for the first forecast of each period while subsequently restart files were used from the previous forecast cycles. We now clarified this in the Model Configuration Section (page 4, lines 16-17).*

3. **P4, L19: Please cite the correct reference of HTAP v2.**

   *The reference has been corrected.*

4. **Table 3: please add the significance level for the correlation coefficient.**

   *For all stations, during both periods, the p-value was found to be lower than 0.05. We have added this in the table description.*

5. **P7 L5: Why the model cannot predict the high NO2 concentration at the site of LIMTRA in both winter and summer? Any emission sources are missing? At the site of LARTRA, the modeled NO2 concentration is underpredicted, any factors would help to improve it?**

   *Traffic stations are located directly adjacent to major roads and are directly influenced by tailpipe emissions from local traffic. As a result, these stations frequently record very high concentrations of pollutants (as shown by the large number of outliers and the large standard deviation), which are not resolved by the atmospheric model that uses a horizontal grid of emissions averaged over 2km x 2km. We have clarified this in the manuscript (page 8, lines 13-15).*

6. **P5 L15 and Table 4: better to explain why the CAMS model results are quite different to the WRF-Chem, what factors may contribute to that, resolution, chemical scheme, or emission?**

   *The results on Table 4 are explained, now with more detail according to the referee's suggestions, for each species separately in paragraphs 3.2, 3.3, and 3.4 for $NO_2$, $O_3$, and PM2.5 respectively.*

7. **P7, L4-25: There shows significant differences in Figure 4 for NO2 forecast between the WRF-Chem and CAMS model for both summer and winter over these sites except Background. Though you have described these differences from the statistic way but did not explain the reasons. What are the major factors contributing to the differences? Are these using the same emission? Or the resolution differences? These need to be investigated.**

   *Due to its short atmospheric lifetime (~1 day), $NO_2$ is detected near the emission sources. CAMS underestimates the $NO_2$ mixing ratios at the residential and traffic stations but captures the background mixing ratios indicates the absence of emission sources in the emission inventory used by the individual CAMS models. In addition, the higher resolution of the emission inventory used by the WRF/Chem model results in better representation of the emission sources, and thus, more accurate $NO_2$ forecasts. More specifically, 1km x 1km emissions were upscaled to the innermost domain resolution of 2km while CAMS uses an inventory of a horizontal resolution of ~6km x 6km, upscaled to 10km. We have updated the text in the manuscript accordingly.*

8. **P8, L2: why CAMS cannot capture the diurnal profile of the wintertime NO2 mixing ratios at the locations with intense anthropogenic activity?**

   *CAMS underestimates the $NO_2$ mixing ratios at the residential and traffic stations, especially during hours with higher anthropogenic activity (traffic) but captures the background mixing ratios. This indicates the absence of local emission sources in the emission inventory used by the individual CAMS models.*

9. **Figure 6: How to explain the big underprediction of CAMS results?**

   *The big underprediction of $NO_2$ by CAMS can be attributed to a large extent to the underestimation of emissions by the anthropogenic emission inventory, since it is more pronounced near the locations with intense anthropogenic activity. $NO_2$ has a short atmospheric lifetime, and it is detected near the emission sources, thus modelling the atmospheric concentrations of $NO_2$ highly depends on the emissions. We have added this in the discussion (page 8, lines 19-24).*

10. **P8, L11, any evidence to the PBL may be related to this issue in WRF-Chem?**

   *During the summer, the boundary layer extends to higher altitudes than in winter. This is reproduced by the WRF/Chem model as shown on Figure S2 added in the Supplement. The*

*boundary layer height is a key factor in the vertical mixing and dilution of near-surface pollutants. Studies have shown that there is a negative correlation between the height of the boundary layer and the concentrations of atmospheric pollutants near the surface (Su et al. 2018, Xiang et al. 2019). We have added these references in the manuscript (page 9, lines 17-19).*

**11. P13, again, the differences for O3 between WRF-Chem and CAMS need to be explained in Fig. 6, especially, the NO2 in CAMS is not good, but the O3 is not too bad, why?**

*The fact that CAMS cannot capture the $NO_2$ mixing ratios can be attributed to missing emission sources in the anthropogenic inventory, as in point 9 above.*

**12. Figure 3: why the O3 is overpredicted in winter, but underpredicted in summer? Just the emission issue? Did you investigate the model performance of Meteorological Fields, which may be related to the O3 production, such as the OH, PBL, etc.**

*O3 is forecast with sufficient accuracy at the background station during both winter and summer (NMB = 7% during winter and 5% during summer). Since the overestimation during both periods is observed near the locations with intense anthropogenic activity, we conclude that this is due to the underestimation in NOx mixing ratios which is a result of less accurate/detailed emissions in the emission inventory.*

**13. P13, L29: why you said that the O3 background concentrations is underestimated? But in Figure 3, the O3 concentration is obviously overestimated over the whole domain.**

*Summertime background $O_3$ concentrations are underestimated by the CAMS model. Figure 3 presents the concentrations from the 1st day forecast by of the WRF/Chem model. We have clarified this in the manuscript and in the figure legend.*

**14. Figure 7: any reasons that may cause the differences between WRF-Chem and CAMS results in the AYMBGR? Which factor cause the overprediction in PARTRA?**

*WRF/Chem is an online coupled model treating meteorological and chemical processes in tandem during the simulations at every timestep, which allows to capture feedbacks between these processes, such as radiation which plays an important role in the production of O3, taking into account all the factors that affect it such as cloud cover and aerosol optical depth. This allows for more accurate representation of the processes that contribute to the O3 tropospheric cycle. The overprediction of $O_3$ at the PARTRA station,*

*can be partly attributed to the underestimation of NOx concentrations (due underestimation of local emissions) and the limited $O_3$ titration by NO.*

**15. P16, L2-L4: please show the precipitation from the model in summer and winter.**

*We have added the modelled precipitation in the Supplement and the relative reference in the manuscript.*

**16. Figure 3: why the PM2.5 is overpredicted in winter, while underpredicted in summer? Why is there increasing PM2.5 concentration during the 3-day period in December 12-14? Is model issue or emission issue?**

*The overprediction shown during the winter is mainly due to increased concentrations during the 3-day period from December 12th to December 14th as can be seen from Figure 8. During these days, increased concentrations of primary PM2.5 aerosols are shown over the whole innermost domain. Primary PM2.5 aerosols, in the MADE mechanism, include the emitted accumulation mode dust particles which account for the 7% of the total dust emissions. The fact that increased concentrations of primary PM2.5 aerosols appear at both background and residential stations indicates that this overestimation is due to overestimation in dust concentrations. This is also supported by the elevated concentrations of coarse dust during this period. Dust is an important aspect of air quality in the broader Eastern Mediterranean and Middle East (MENA) region and the model ability to simulate aeolian dust concentrations and the processes dust undergoes in the atmosphere affects episodic model skill. We have added this in the discussion (page 18, lines 25-34).*

**17. Figure 6: Both two models cannot capture the PM2.5 diurnal cycle, can you explain the reason?**

*The increased observed PM2.5 concentrations during the evening hours at these stations (Figure 6) are attributed to emission sources from fossil fuels combustion used for household heating. These sources are not properly captured by the emission inventory resulting in the underestimation of PM2.5 concentrations, as discussed in Georgiou et. al. (2020). We now highlight this in the manuscript (page 18, lines 20-21).*

**18. Figure 8: the model results cannot capture most of the peaks at the sites of LARTRA, NICTRA, PARTRA, can you explain why?**

*Traffic stations are located directly adjacent to major roads and are directly influenced by tailpipe emissions from local traffic. As a result, these stations frequently record very high concentrations of pollutants (as shown by the large number of outliers and the large standard deviation), which are not resolved by the atmospheric model that uses a horizontal grid of emissions averaged over 2km x 2km.*

**References**

Gupta, M. and Mohan, M.: Validation of WRF/Chem model and sensitivity of chemical mechanisms to ozone simulation over megacity Delhi, Atmos. Environ., 122, 220–229, doi:10.1016/j.atmosenv.2015.09.039, http://dx.doi.org/10.1016/j.atmosenv.2015.09.039, 2015.

Balzarini, A., Pirovano, G., Honzak, L., ??abkar, R., Curci, G., Forkel, R., Hirtl, M., San Jos??, R., Tuccella, P., and Grell, G. A.: WRF-Chem model sensitivity to chemical mechanisms choice in reconstructing aerosol optical properties, Atmos. Environ., 115, 604–619, doi:10.1016/j.atmosenv.2014.12.033, 2015

Im, U., Bianconi, R., Solazzo, E., Kioutsioukis, I., Badia, A., Balzarini, A., Bar??, R., Bellasio, R., Brunner, D., Chemel, C., Curci, G., Flemming, J., Forkel, R., Giordano, L., Jim??nez-Guerrero, P., Hirtl, M., Hodzic, A., Honzak, L., Jorba, O., Knote, C., Kuenen, J. J. P., Makar, P. A., Manders-Groot, A., Neal, L., P??rez, J. L., Pirovano, G., Pouliot, G., San Jose, R., Savage, N., Schroder, W., Sokhi, R. S., Syrakov, D., Torian, A., Tuccella, P., Werhahn, J., Wolke, R., Yahya, K., Zabkar, R., Zhang, Y., Zhang, J., Hogrefe, C., and Galmarini, S.: Evaluation of operational on-line-coupled regional air quality models over Europe and North America in the context of AQMEII phase 2. Part I: Ozone, Atmos. Environ., 115, 404–420, doi:10.1016/j.atmosenv.2014.09.042, 2015

Mar, K. A., Ojha, N., Pozzer, A., and Butler, T. M.: Ozone air quality simulations with WRF-Chem (v3.5.1) over Europe: Model evaluation and chemical mechanism comparison, Geosci. Model Dev., 1, 1–50, doi:https://doi.org/10.5194/gmd-9-3699-2016, 2016

Georgiou, G. K., Christoudias, T., Proestos, Y., Kushta, J., Hadjinicolaou, P., and Lelieveld, J.: Air quality modelling in the summer over the Eastern Mediterranean using WRF / Chem : Chemistry and aerosol mechanisms intercomparison, Atmospheric Chem. Phys., pp. 1–25, doi:https://doi.org/10.5194/acp-18-1555-2018, 2017.

Su, T., Li, Z., and Kahn, R.: Relationships between the planetary boundary layer height and surface pollutants derived from lidar observations over China: Regional pattern and influencing factors, Atmospheric Chemistry and Physics, 18, 15 921–15 935, doi:10.5194/ACP-18-15921-2018, 2018.

Xiang, Y., Zhang, T., Liu, J., Lv, L., Dong, Y., and Chen, Z.: Atmosphere boundary layer height and its effect on air pollutants in Beijing during winter heavy pollution, Atmospheric Research, 215, 305–316, doi:10.1016/J.ATMOSRES.2018.09.014, 2019.

Georgiou, G. K., Kushta, J., Christoudias, T., Proestos, Y., and Lelieveld, J.: Air quality modelling over the Eastern Mediterranean: Seasonal sensitivity to anthropogenic emissions, Atmos. Environ., 222, 117 119, doi:10.1016/j.atmosenv.2019.117119, 2020.